# The Crohn's disease-related bacterial strain LF82 assembles biofilm-like communities to protect itself from phagolysosomal attack

Victoria Prudent[1], Gaëlle Demarre[1], Emilie Vazeille[2], Maxime Wery [3], Nicole Quenech'Du[1], Antinéa Ravet [1], Julie Dauverd - Girault[1], Erwin van Dijk[4], Marie-Agnès Bringer[2,5], Marc Descrimes[3], Nicolas Barnich[2], Sylvie Rimsky[1], Antonin Morillon [3] & Olivier Espéli [1✉]

Patients with Crohn's disease exhibit abnormal colonization of the intestine by adherent invasive *E. coli* (AIEC). They adhere to epithelial cells, colonize them and survive inside macrophages. It appeared recently that AIEC LF82 adaptation to phagolysosomal stress involves a long lag phase in which many LF82 cells become antibiotic tolerant. Later during infection, they proliferate in vacuoles and form colonies harboring dozens of LF82 bacteria. In the present work, we investigated the mechanism sustaining this phase of growth. We found that intracellular LF82 produced an extrabacterial matrix that acts as a biofilm and controls the formation of LF82 intracellular bacterial communities (IBCs) for several days post infection. We revealed the crucial role played by the pathogenicity island encoding the yersiniabactin iron capture system to form IBCs and for optimal LF82 survival. These results illustrate that AIECs use original strategies to establish their replicative niche within macrophages.

[1] CIRB – Collège de France, CNRS-UMR7241, INSERM U1050, PSL Research University, Paris, France. [2] Microbes, Intestin, Inflammation et Susceptibilité de l'Hôte. UMR Inserm/ Université de Clermont -Auvergne U1071, USC INRA 2018, Clermont, Ferrand, France. [3] ncRNA, Epigenetic and Genome Fluidity, Institut Curie, Sorbonne University, CNRS UMR 3244, Paris, France. [4] Next-Generation Sequencing Service - I2BC, I2BC-CNRS, Gif-sur-Yvette, France. [5] Centre des Sciences du Goût et de l'Alimentation, AgroSup Dijon, CNRS, INRA, Université Bourgogne Franche-Comté, Dijon, France. ✉email: olivier.espeli@college-de-france.fr

Patients with Crohn's disease exhibit abnormal colonization of the intestine by proteobacteria, among which the adherent invasive *E. coli* (AIEC) family has been characterized. AIEC are found in intestinal lesions of patients with inflammatory bowel disease. These bacteria are found mainly in the mucus, they adhere to epithelial cells and colonize them to survive inside macrophages. The mechanisms underlying AIEC adaptation for survival and growth inside macrophages are not yet fully understood. Previous work performed with murine macrophage cell lines has revealed that the prototype AIEC strain LF82 proliferates in a vacuole exhibiting the characteristics of a mature phagolysosome[1,2]. In such an environment, AIEC should encounter acidic, oxidative, genotoxic, and proteic stress. Screening of genes involved in LF82 fitness within macrophages has revealed that the HtrA, DsbA, or Fis proteins are required for optimum fitness[3–5]. These observations confirmed that LF82 encounters stress in phagolysosomes. However, much remains to be learned about the host–pathogen interactions that govern AIEC infection biology. AIEC expresses various virulence factors that might play roles in digestive tract colonization, adherence and invasion of epithelial cell, and intracellular survival in phagolysosomes. The diversity of virulence factors displayed by multiple AIEC strains suggests that members of this pathovar may have evolved different strategies to colonize their hosts[6].

We have recently demonstrated that the stringent response (the bacterial checkpoint involved in dealing with nutrient limitation) and SOS response (the bacterial pathway involved in DNA repair) are critical for AIEC survival and multiplication within macrophages[7]. The triggering of these two stress pathways has direct consequences, creating heterogeneity in the bacterial population with replicating and non-replicating bacteria, that contribute respectively to increase population size and tolerate the stress. The non-replicative population of LF82 tolerates antibiotics. Such bacteria are called persisters and are suspected to be the cause of relapsing infectious chronic diseases, for instance in tuberculosis or cystic fibrosis.

In addition to persistence, biofilm structures represent another source of antibiotic tolerance. Biofilms are communities of cells attached to surfaces and held together by a self-produced extracellular matrix. The matrix is composed of various extracellular DNA molecules, proteins, and polysaccharides, depending on the bacterial species[8]. Cells in the biofilm state exhibit increased protection against desiccation and harmful substances, including antibiotics and the host immune response molecules[9]. Biofilms retain bacteria to diverse surfaces, including, in the case of certain pathogens, tissues and, less frequently, intracellular surfaces. Intracellular biofilms, also called intracellular bacterial communities (IBCs), have only been described when uropathogenic *E. coli* (UPEC) invade urothelial cells lining the urinary bladder[10]. This biofilm allows to escape host defense and UPEC persist despite antibiotic therapy[11].

In the present work, we investigated the mechanism by which the AIEC pathobionts such as the LF82 strain forms microcolonies inside macrophage phagolysosomes. We found that LF82 microcolonies form IBCs with an extracellular matrix composed of exopolysaccharides, and curli fibers surrounding each bacterium. Among the list of critical genes for the development of *E. coli* biofilms, we revealed the specific pathway involved in IBCs formation and survival in macrophages. Finally, by combining genomic screens (dual RNA-seq and TN-seq) we unveiled an LF82's pathogenicity island required for IBCs. This pathogenicity island, called HPI, allows iron providing to the IBC. Interestingly our results unveil a tri partite link between iron homeostasis and biofilm matrix production for intracellular pathogen proliferation. HPI is present in many AIEC and pathogen bacteria, suggesting that the strategy developed by LF82 might be common for

AIEC, and other facultative intracellular pathogens and pathobionts that survive in the particular niche of the phagolysosome.

## Results

### The AIEC strain LF82 forms intracellular bacterial communities (IBCs) within phagolysosomes.
In the minutes following phagocytosis, the induction of a stringent response blocks bacterial cell division and curbs the expansion of the LF82 population. This 6–10 h step can lead to the formation of LF82 persisters[7]. When LF82 resumes growth, the expansion of the population is dependent on the ability of the bacteria to repair lesions (e.g., DNA lesions). Bacteria involved in this multiplication stage divide 4–6 times in 10 h. This growth phase leads frequently to the formation of vacuoles containing more than 20 bacteria within human THP1 macrophages derived from monocytes and up to hundreds of bacteria in murine Raw 264.7 macrophage cell lines (Fig. 1a, b). To discriminate whether the formation of large vacuoles corresponded to only the clonal multiplication of one or few phagocytic bacteria or, alternatively, to the fusion of different vacuoles, Raw macrophages were infected successively (1 h interval) with GFP-tagged LF82 and mCherry-tagged LF82 (Fig. 1c). One hour of post-infection (P.I.) with LF82-mCherry, most vacuoles contained only one type of LF82 (either green or red), while 24 h P.I., a small proportion of vacuoles presented both types of LF82 (Fig. 1c, low panel). Surprisingly, we observed that red and green LF82 formed clonal sectors within vacuoles. These observations suggest that the formation of large vacuoles is not the consequence of multiple fusion events, and that LF82 is not free to move within a vacuole. To directly measure the ability of LF82 to move inside a vacuole, we used fluorescence recovery after photobleaching (FRAP) experiments. Bleaching of several spots inside large vacuoles did not lead to recovery of the fluorescence and demonstrated strong adherence to LF82 (Fig. 1d, e). We deduced that limited movement of LF82 promoted the formation of colonies within phagolysosomes, akin to the IBCs observed for UPEC within bladder cells[10].

### Transcriptomic analysis of LF82 survival within macrophages.
We performed a dual RNA-seq experiment[12] of THP1 macrophages infected by LF82 at 1 and 6 h of P.I. to characterize the transcriptional program corresponding to the formation of IBCs. Experiments were performed in duplicate; about 4 million bacterial and 300 million human reads per sample were collected (Supplementary Data 1) and DEseq P-values were calculated for each gene (Fig. 2a, b). We first analyzed the bacterial transcriptome. The transcriptomic response of LF82 during macrophage infection was considerable; the expression of 700 and 1000 genes was significantly changed at 1 and 6 h of P.I., respectively (Fig. 2a and Supplementary Data 2). At 1 h of P.I., a majority of the genes were downregulated, while at the 6 h, both upregulated and downregulated genes were detected. Genes involved in the responses to the toxic phagolysosome's environment were among the most upregulated, while genes involved in the carbon metabolism and bacterial motility were the most down regulated (Fig. 2a). COG analysis confirmed this observation revealing that the main upregulated pathways were the response to external stimuli (particularly pH variation and SOS response), biofilm regulation, cell communication, and organophosphate metabolic processes (Supplementary Data 3). The main downregulated pathways were metabolic pathways, including energetic metabolism, nucleoside phosphate metabolism, and carbohydrate metabolism (Supplementary Data 3). To get a global view of the transcriptomic response, we analyzed the most significantly changed 500 genes at 1 h (Supplementary Fig. 1a) and 6 h of P.I.

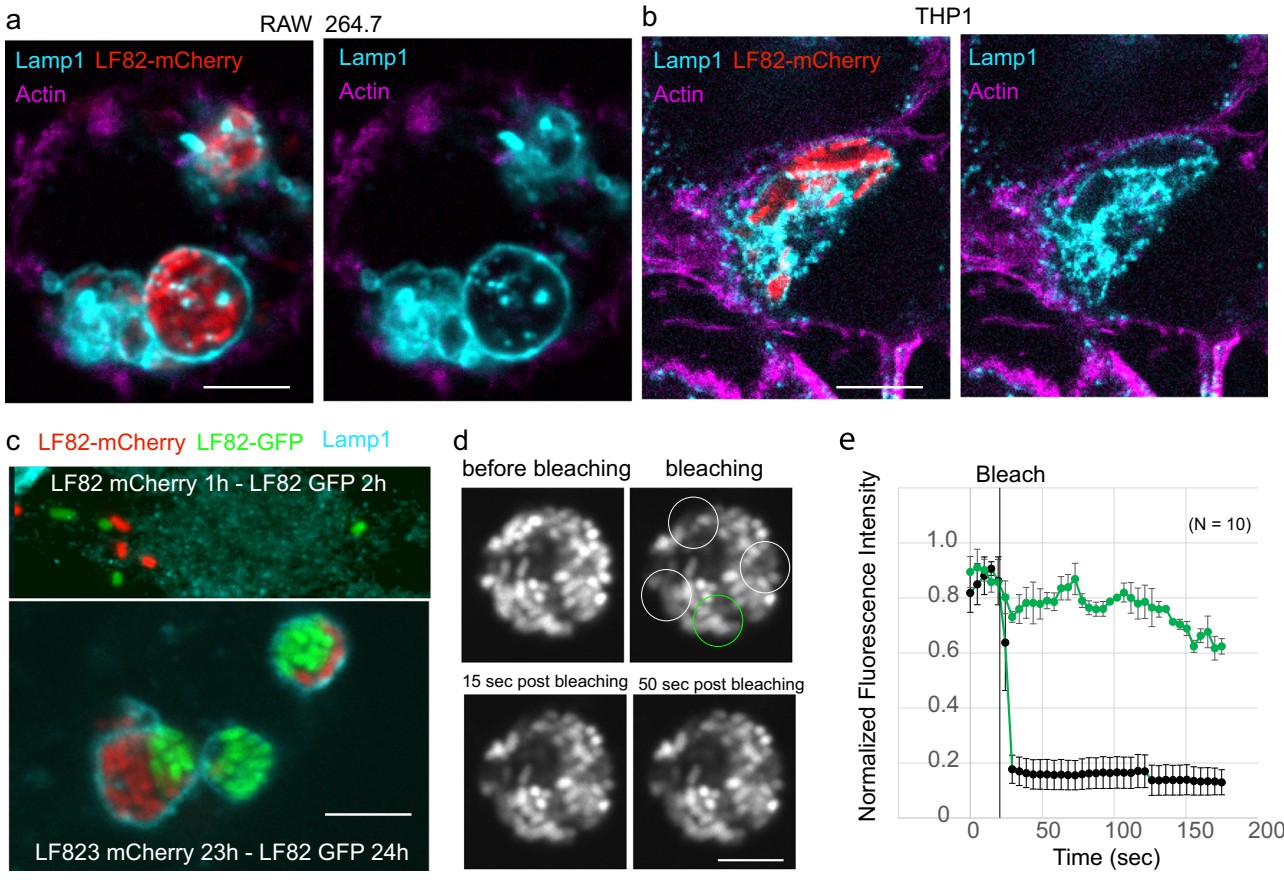

**Fig. 1 Intracellular LF82 form intracellular bacterial communities inside phagolysosomes. a** Imaging of the phagolysosomes (antibody for Lamp1, cyan) of Raw macrophages (actin labeling with phalloidin, magenta) infected by LF82-mCherry (red) at an MOI of 30 at 24 h of P.I. **b)** Imaging of the phagolysosomes of THP1 macrophages infected by LF82-mCherry at an MOI of 30 at 24 h of P.I. **c** Coinfection experiment. Raw macrophages were infected first with LF82-mCherry, treated with gentamycin for 1 h to remove free LF82-mCherry and subsequently infected with LF82-GFP. At 24 h of P.I., phagolysosomes were labeled with the Lamp1 antibody (cyan) and imaged. **d** FRAP experiments on phagolysosomes containing LF82-GFP at 24 h of P.I. White circles outline the bleached regions, and green circles outline the control region. The scale bars are 5 μm. **e** Fluorescence quantification of the FRAP experiment presented in **d**. Data are average +/− SD of 10 FRAP and control areas.

(Supplementary Fig. 1b). We grouped genes in 22 categories based on available annotations. Some categories were densely populated (Central metabolism, for example, present nearly 100 genes with significant mRNA fold change at 1 or 6 h of P.I.) and some contained only few representatives (three efflux pumps). This analysis gave an image of the growth conditions encountered by AIEC LF82 within macrophages. It confirmed the presence of acid and genotoxic stresses, and showed: i) the alteration of the bacterial envelope (membrane, cell wall, and periplasm), ii) the reduction of protein biosynthesis and cell cycle activity, presumably linked to stringent response induction[7], iii) the switch from motile to biofilm behavior and, iv) the global reduction of catabolic and energetic metabolism (Fig. 2c). We tested the robustness of these transcriptomic responses with RT-qPCR and dual RNA-seq experiments performed during Raw 264.7 cells infection (Supplementary Fig. 2). RT-qPCR confirmed the induction of the biofilm pathway in Raw264.7 macrophage at 6 h of P.I., and its maintain at later time points (Supplementary Fig. 2b). RNA-seq in Raw 264.7 cells was not fully successful, we collected low amount of sequencing reads mapping to the LF82 genome, this prevented statistical analysis. However, the combination of the two replicates allowed us to observe an overall good correlation of gene expression with the THP1 data (Pearson coefficient, $P = 0.75$, Table S2, SI S2B) and the regulation of similar pathways (see below). We inspected the THP1's infection RNA-seq according to the E. coli K12 regulons described in the

Regulon Database[13]. Among the upregulated regulons, those associated with the response to acidic pH are largely over-represented (Supplementary Fig. 1c), and genes involved in the response to acidic pH are highly overexpressed as early as 1 h of P.I. (Supplementary Fig. 1d). Members of the second family of upregulated regulons govern the bacterial growth mode (cell division, LPS, capsule, or biofilm determinants) (Supplementary Fig. 1c). Among the genes controlled by these TFs, we observed, upon entry into the macrophage, a general upregulation of the genes annotated for biofilms and adhesion (Fig. 2d). Consistent with this change in the LF82 growth, upon entry into the macrophage, we observed a severe downregulation of the FlhDC regulon, which contains motility genes (Supplementary Fig. 1d). Biofilm formation frequently corresponds to a switch toward a slower metabolism, which is exactly what the RNA-seq data suggested, with downregulation of genes involved in glycolysis and ATP production (Supplementary Fig. 1d). We mapped RNA-seq results on the KEGG ko2026 pathway for biofilm formation of commensal *E. coli* (Fig. 2d). This map suggests that the lack of nutrient, perhaps some amino acids, induces stringent response and consequently represses the expression of FlhDC, therefore flagellar assembly could be stopped and bacteria become non-motile. At the same time acid, osmolarity and envelope stresses could lead to the induction of *csgD* via OmpR and RpoS. CsgD induces the expression of the operons encoding curli fibers. Quorum sensing, via the BarA/UvrY two component system, the

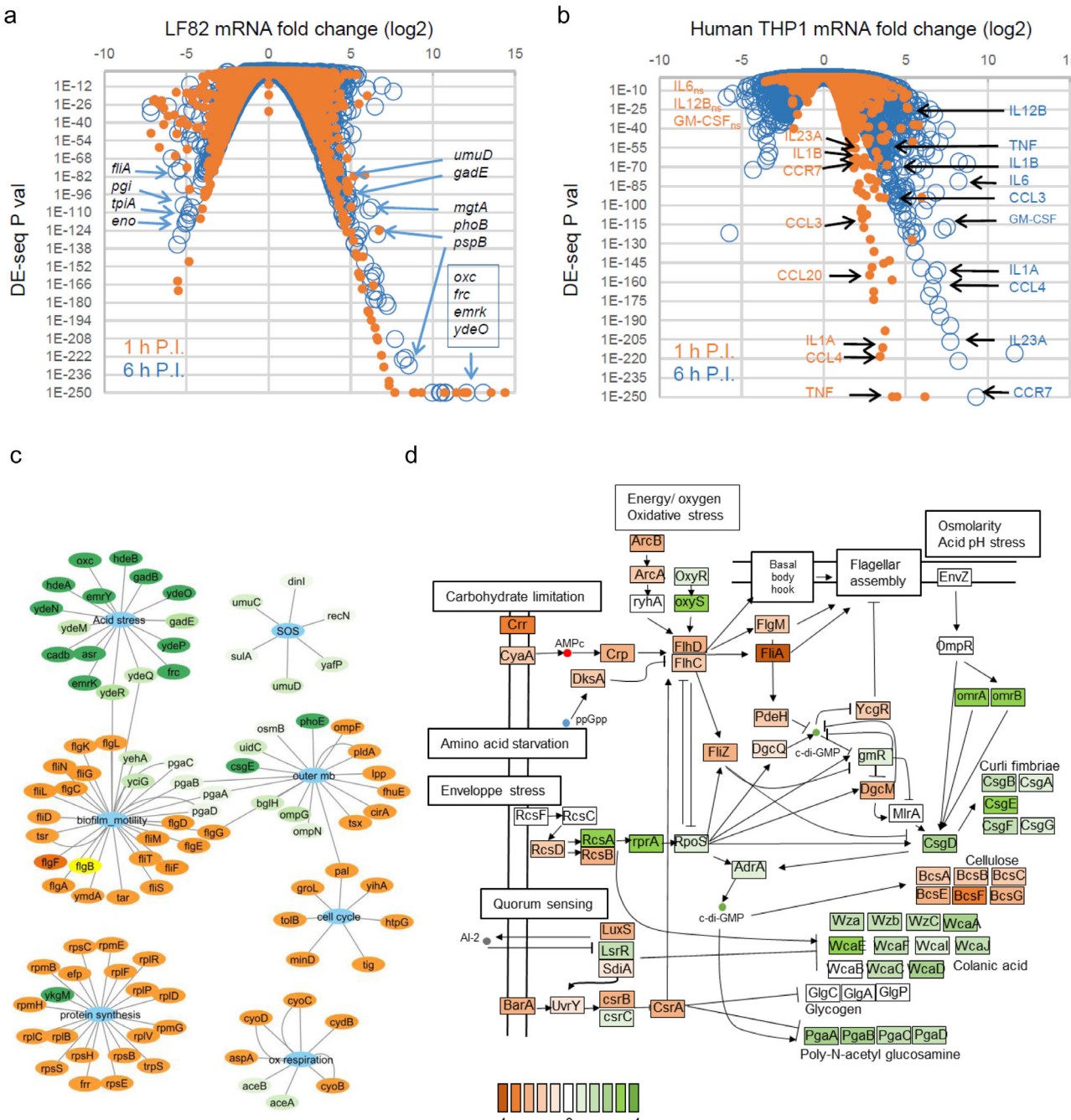

**Fig. 2 Transcriptomic analysis of macrophage infection by LF82. a** Expression data for the LF82 genes during infection of THP1 macrophages. Selected genes with highly significant expression changes at 6 h of P.I. are noted. **b** Expression data for the human genes from RNA-seq experiments at 1 and 6 h of P.I. Gene markers of the M1 pro inflammatory macrophage phenotype are noted. **c** Representation of the main *E. coli* pathways presenting large transcriptomic responses at 6 h of P.I. of THP1 macrophages (top 500 genes ranked on DE-seq). These networks were constructed with cytoscape and were based on the data presented on Supplementary Fig. 1. Upregulated genes (green) and downregulated genes (orange). **d** Annotation of the biofilm formation pathways adapted from KEGG ko2026 according to RNA-seq data at 6 h of P.I. vs. planktonic exponential growth.

LuxS/LsrR system and the CsrA repression will lead to the expression of the operons involved in the production of poly-N-acetyl glucosamine and colanic acid. Genes involved in the production of cellulose are repressed; the available literature does not allow to explain this observation. Finally, RNA-seq data of the Raw 264.7 macrophage's infection confirmed the global induction of the biofilm pathway at 6 h of P.I. (Supplementary Fig. 2c). These transcriptomic observations strongly suggest that LF82 switches from a planktonic to a biofilm mode of growth once within macrophage, and suggest that an extracellular matrix

is responsible for the observed adherence and clonal growth within IBCs.

**Transcriptomic response of the macrophages to AIEC infection.** We analyzed the Dual RNA-seq data to monitor macrophage response to LF82 infection. LF82 infection strongly impacted the THP1 macrophage transcriptome (Fig. 2b and Supplementary Data 6). As expected, we observed the induction of the antimicrobial humoral immune response (the CXCL1,

CXCL2, CXLC3, IL1, and IL8 genes were among the most significantly affected genes); the inflammatory response (TNF, CCR7, NFKBIA, OSM, CCL3L3, NFBIZ, and PTSG2); TNF and the response to its production (the CCL1, CCL3, CCL3, and CCL20 genes); and the lymphocyte activation pathway (CD80 and IL23A). The response is stronger (i.e., more genes significantly changed, increased number of COG represented in the list of significantly changed genes and higher mRNA fold ratio) at 6 h of P.I. compared to 1 h. P.I. Upon infection macrophages become polarized, this is frequently illustrated by the presence of two phenotypes: M1 like macrophages that are pro inflammatory and M2 like macrophage that are anti inflammatory. Recent work demonstrated that *Salmonella* is able to drive polarization of mice BMM macrophages toward an M2 more permissive state[14]. In human, the separating line between M1 and M2 like macrophages is rather represented by a continuum where boundaries are still unclear. However several transcriptomic markers were attributed to M1 (IL12B, CCR7, IL1A, IL1B, IL23A, TNF, CCL3, CCL4, and GM-CSF) and M2 (GM-CSF, IL4, IL10, IL13, CCL1, CCl2, CCL17, CCL18, CCL22, CCL23, CCL24, CCL26, IL1R, and TGFB)[15]. RNA-seq data showed that most M1 markers are significantly overexpressed upon LF82 infection (Fig. 2b) while none of the M2 markers were upregulated (Supplementary Data 6). This confirmed that LF82 encounter stressful environment during its first hour of survival within pro-inflammatory macrophages.

**IBCs are structured by an extracellular matrix.** Using fluorescent lectins to visualize the exopolysaccharide matrix, we detected Wheat germ agglutinin (WGA) and soybean agglutinin (SBA) around bacteria in the phagolysosomes (Fig. 3a and SI S3A). Labeling of WGA and SBA revealed that both form envelopes surrounding each bacteria. WGA labeling was weak at 1 h of P.I, while it was clearly detected in Raw 264.7, THP1, and HDMM at 24 h of P.I. (Supplementary Fig. 3b–d). In addition, we also frequently observed a strong labeling of bacterial periphery that colocalized with Lamp1. WGA and SBA were also detected in many other organelles of the macrophage including Golgi apparatus, that contains glycosylated proteins and lipids. To challenge the specificity of the WGA staining, we performed coinfection with a K12 C600 *E. coli* strain that is not as efficient as LF82 to proliferate within macrophages, making rare occasional foci consisting of several bacteria (Fig. 3b). In this assay, WGA labeling was strongly reduced in the C600 vacuoles compared to the LF82 vacuoles (Fig. 3b, c). We used STED super-resolution microscopy to refine our analysis of the WGA staining in WT LF82 and LF82 lacking the regulator *csgD* and the *pgaABCD* operon encoding the exopolysaccharide component of the matrix. WGA labeling around bacteria inside the phagolysosomes was observed for the WT strain while no WGA labeling was detected around LF82 Δ*csgD* and LF82Δ*pgaABCD* (Fig. 3d). All together, these observations confirmed that exopolysaccharides observed around LF82 are from bacterial origin. The absence of an exopolysaccharide matrix in LF82Δ*pgaABCD* was expected, because this operon directly encodes the enzymes dedicated to the synthesis and export of exopolysaccharide; however, the connections between CsgD and the exopolysaccharide matrix have not been documented in commensal *E. coli*[16]. Prolonged infection experiments showed that WGA labeling was present in the IBC formed by LF82 up to 72 h of P.I., but absent in LF82 Δ*csgD*. This suggest that exopolysaccharide production is abolished rather than just delayed in the *csgD* mutant (Supplementary Fig. 3e). Thus, we also concluded that CsgD is an important factor to build LF82-specific matrix within macrophage. Because CsgD is the main controller of the production of curli fibers, we analyzed curli by immunostaining with antibodies against CsgA or CsgB[17];

unfortunately, we could not observe labeling inside macrophages with antibodies. Therefore, we tested the antibodies on bacteria recovered from lysed macrophages 24 h P.I., LF82 showed strong CsgA labeling at its periphery, while C600 presented much weaker labeling (Fig. 3e, f). The genes allowing the biosynthesis and export of colanic acid, another exopolysaccharide (*wza-c, wcaA-L*), were also expressed in the phagolysosomes (Fig. 2d). However, we did not detect any specific labeling of LF82 vacuoles with Concanavalin A (ConA) or Peanut Agglutinin (PNA), that reveals β galactosides present in colanic acid (Supplementary Fig. 4). The genes encoding the protein responsible for cellulose biosynthesis and export were downregulated in the macrophage (Fig. 2d), suggesting that cellulose is not involved in LF82 IBC. Our results demonstrate that LF82 produce a biofilm-like matrix containing at least poly-β-1,6-*N*-acetyl-D-glucosamine exopolysaccharide and amyloid fibrils composed of curlin to organize IBCs.

**The extracellular matrix affects IBC size and LF82 survival within macrophages.** To characterize better the mechanism by which biofilm-like matrix involved in IBC, we constructed deletion mutants lacking genes controlling extracellular matrix production (*csgD, adrA, dgcE, pdeH, ompR, rcsCBD*), genes encoding for structural components of the matrix (*pgaA, wza, waaWVL, bcsB, glgCAP*), genes involved in surface adhesion (*fimH*) and quorum sensing (*qseBC*). We tested the ability of these mutants to form biofilms on the abiotic surface of 96-well polystyrene plates. Most single mutants presented a clear defect to form biofilms (Supplementary Fig. 5a). We then tested some of these mutants for their ability to survive for 24 h within Raw 264.7 macrophages, THP1 macrophages, and human monocyte-derived macrophages (HMDMs) from blood (Fig. 4a–c). In Raw macrophages, viability was reduced by 20–60% for most of the mutants. In THP1 macrophages, viability was reduced but to a lesser extent. Finally, in the HMDMs, the viability of the *csgD* mutant was also reduced. One important aspect of LF82 survival within macrophages is the formation of nongrowing bacteria immediately upon infection and control of the stringent response[18]. IBC are formed during the later replicative phase, starting 10 h of P.I., we observed that only 4–10% of the vacuoles contained more than 16 bacteria at 24 h of P.I. In addition, many LF82 were included in small vacuoles with less than eight bacteria (Fig. 4d, e). In contrast, using the *rcsBD, pgaA, csgD, waaWVL, wza* mutants we observed that, in average, vacuoles contain less bacteria. This suggests that when the biofilm matrix is altered the proportion of LF82 able to form or to maintain large IBC structures is affected. Altogether, these results show that IBCs constitute an important determinant of the LF82 infection program.

**Chemical inhibition of biofilm matrix formation affects the formation of IBC and LF82 survival.** To further test the impact of IBCs on the survival of LF82, we used cis-2-decenoic acid, a well-described inhibitor of biofilm formation and stability. We observed a reduction in WGA intensity around LF82, suggesting a reduction in the amount of exopolysaccharide in the matrix (Fig. 4f), and concomitantly, we observed a reduction in LF82 survival within macrophages (Fig. 4g). The survival of the LF82 *csgD* mutant was not affected by cis-2-decenoic acid, confirming that the drug targets the biofilm matrix. Altogether, our results demonstrate that the IBC structure formed within phagolysosomes is beneficial for LF82 survival in macrophages.

**LF82 genes involved in macrophage survival.** To monitor the formation of the extracellular matrix and design mutants for

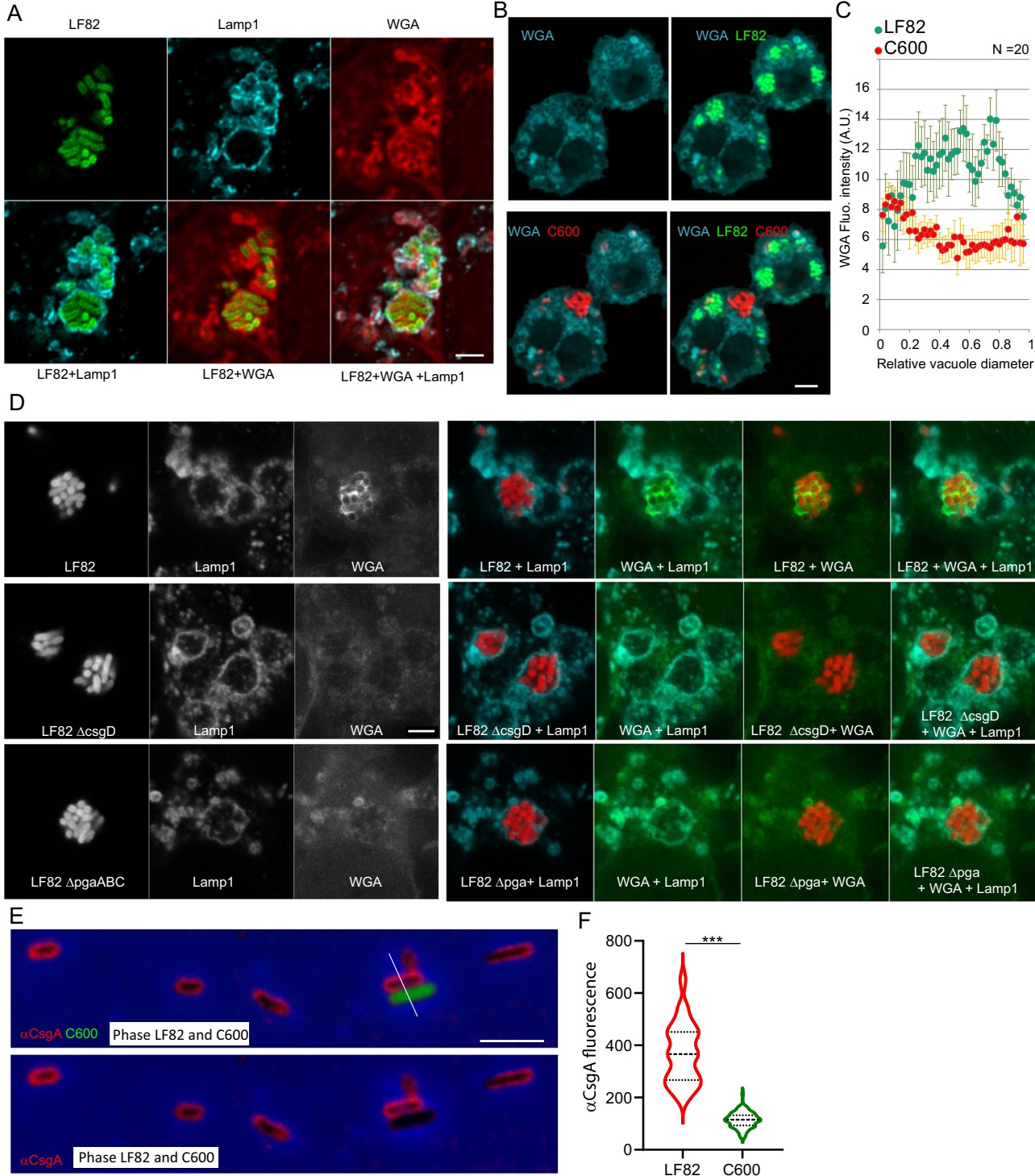

**Fig. 3 Intracellular LF82 IBCs present a biofilm-like matrix. a** Imaging of extracellular polysaccharide structures revealed by wheat germ agglutinin (WGA) labeling around LF82-GFP (green) inside phagolysosomes (antibody for Lamp1, cyan) of Raw macrophages. 24 h of P.I. **b** At 24 h of P.I. polysaccharide structures are observed inside phagolysosomes containing LF82-GFP but not inside phagolysosomes containing only C600-mCherry bacteria. **c** Quantification of the WGA fluorescence intensity along the vacuole diameter of raw macrophages infected by LF82 or C600 for 24 h of P.I. Values represent median $+/-$ SD of 20 vacuoles. **d** STED superresolution imaging of the extracellular polysaccharide structures revealed by WGA staining around LF82 IBCs formed by the WT and the *csgD* and *pgaABC* mutants. **e** Imaging of curli (antibody for CsgA, red) formed around LF82 during macrophage infection. Macrophages infected with LF82 and C600-GFP were lysed with Triton at 24 h of P.I. before fixation. **f** Quantification of the fluorescence intensities along the cross section of LF82 and C600 (white line drawn in **e**). $N = 100$ bacteria of a representative experiment, the data were analyzed using Student's *t*-test: ***$P < 0.001$. The scale bars are 5 μm.

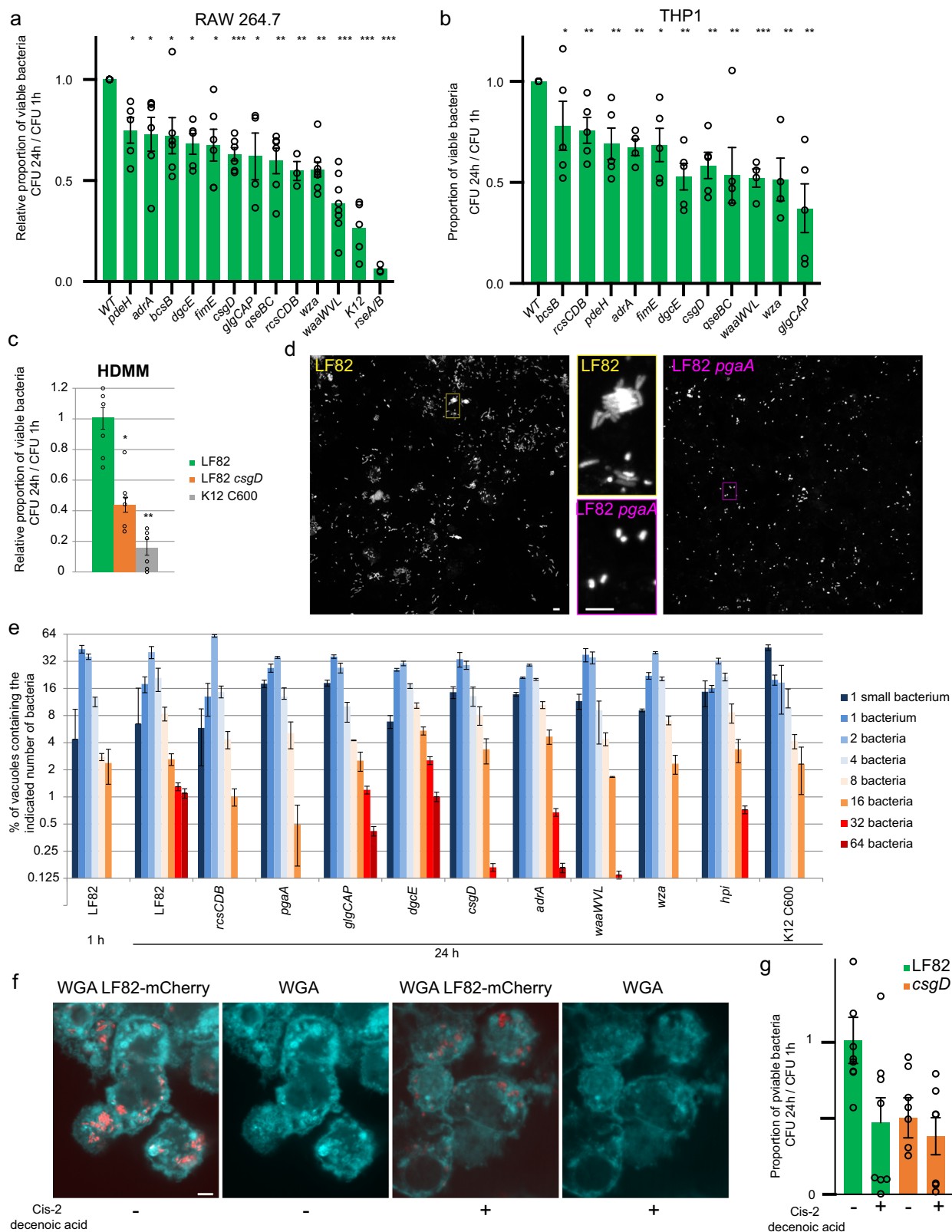

viability assays, we used the extensive knowledge available in the literature on biofilms of laboratory *E. coli* strains. However, we observed a clear difference in matrix formation and survival between LF82 and K12. These observations suggest that some of the LF82 specific genes or mutations might be responsible for its higher biofilm formation capacity, and its ability to form IBC within macrophages. To identify them, we performed transposon

mutagenesis and analysis by whole genome sequencing, Tn-seq[19] (Fig. 5a). Knowing that, in average, vacuoles contain 4–5 times more LF82 in the Raw 264.7 macrophages compared to the THP1 cell line, we performed this experiment with the Raw cell line. Preliminary competitions experiments (Supplementary Fig. 5b) showed the limited impact of known mutants affecting the growth phase of LF82 within macrophages after 24 h of infection;

**Fig. 4 Deletions of the genes involved in the synthesis of the extracellular matrix curb LF82 survival. a** Proportion of viable bacteria at 24 h of P.I. in Raw macrophages in comparison to those present at 1 h. LF82 and LF82 deletion mutants (green) and K12-C600 were infected at an MOI of 30. Values represent the average of 3–7 experiments ± SEM. The data were analyzed using Student's *t*-test: *$P < 0.05$, **$P < 0.01$, ***$P < 0.001$, ****$P < 0.0001$.
**b** Proportion of viable bacteria at 24 h of P.I. of THP1 macrophages in comparison to those present at 1 h. The data were analyzed as in **a**. **c** Proportion of viable LF82, LF82 *csgD* and K12-C600 bacteria at 24 h of P.I. in human derived macrophages from blood monocytes (HDMMs) in comparison to those present at 1 h. The data were analyzed as in **a**. **d** Imaging of the IBCs formed at 24 h of P.I. with LF82-mCherry and the LF82 *pgaA* mutant. **e** Distribution of IBC size in the population of the WT LF82 strain and the *rcsBD, pgaA, glgCAP, dgcE, csgD, adrA, waaWVL, wza, hpi* mutants and the K12 C600 strain. The scale bars are 5 µm. **f** Inhibition of exopolysaccharide matrix synthesis by the addition of cis-2-decenoic acid at 6 h of P.I.; imaging was performed at 24 h of P.I. **g** CFU of LF82 and the LF82 *csgD* mutant in the presence of cis-2-decenoic acid (1 µM) in the medium for 6 h from 18 to 24 h. Values represent the average of 3–7 experiments ± SEM.

however, this impact became obvious after three successive rounds of infection and LB outgrowth. We, therefore, performed three rounds 24 h infections; we allowed a 24 h in LB regrowth step in between each infection. The density of transposon insertion per genes was measured for the population that went through macrophages (3 × 24 h in macrophage + 3 × 24 h in LB) and kept in LB (3 × 24 h in LB) (Fig. 5a). Our two replicates of the whole experiment were in good agreement (Fig. 5a). We used fold change of the average insertion density for further analysis (Fig. 5b, c). Approximately 180 genes were identified that present a normalized insertion density >2 in the library (i.e., genes that are not essential for LF82 in LB) and a reduced fold change in the selection experiment (Fig. 5b and Supplementary Data 4). This group is enriched in transcription factors and phospholipid transport (Supplementary Data 5). The veracity of our analysis pipeline was supported by findings at the top of the hit list of previously validated important genes for LF82 survival within macrophages (*dksA, rpoS, slyB, degP, pspB-C-F, phoP*, Fig. 5b and Supplementary Data 4). We observed only weak Tn-seq signals for the genes encoding for proteins involved in the formation of the biofilm matrix (Supplementary Data 4). However, we observed strong essentiality signals for regulators of the biofilm pathway (*rpoS, dksA, barA, uvrY, rcsB, proQ, comR, bhsA*) (Fig. 5b). The lack of genes involved in the building of the exopolysaccharide and curli matrix suggest a certain redundancy in between matrix components or non-cellular autonomous effects allowing mutated bacteria to be protected by the matrix produced by WT LF82 present in the same vacuole. We combined RNA-seq and Tn-seq data to screen for LF82 specific genes putatively involved in IBC growth or survival (Fig. 5c). Three gene clusters emerged from this analysis: the pathogenicity island II (also called HPI, in other pathogens), a putative type 6 secretion system (T6SS) and a putative carbohydrate's metabolism gene cluster. HPI expression was induced at 6 h of P.I. but not at 1 h of P.I. (Fig. 5d), a timing corresponding with the appearance of robust exopolysaccharide staining (Supplementary Fig. 3). In contrast, overexpression of the phosphoglucoside and T6SS clusters was observed as early as 1 h of P.I. (Supplementary Fig. 6). We, therefore, focused our analysis on the HPI, the function of which in *Yersinia pestis* has been well characterized[20]. HPI allows the production of a siderophore called Yersiniabactin, its export and its reentry into the bacteria when iron has been captured. Yersiniabactin is a small peptide synthesized by non-ribosomal peptide synthases encoded on the island (*irp1* and *irp2* genes). Tn-seq data revealed that insertions in *ybtX, irp1, irp2, ybtE,* and *fyuA* genes were counter selected in the first round of infection, but not when LF82 was subjected to three rounds of selection (Fig. 5d). In contrast, the *ybtS, ybtP, ybtQ, ybtT,* and *ybtU* genes were still under selective pressure after three rounds of infections (Fig. 5d). A possible explanation for these observations is that Yersiniabactin production and import is advantageous during infection, but might be detrimental during the recovery periods in LB between infections.

**Iron scavenging by yersinibactin determines LF82 fate within macrophage.** HPI is composed of two divergent open reading frames controlled by a single AraC-like regulator (*ybtA*, LF82_p299). We obtained a mutant in which the first part of the island from *ybtS* to *irp1* was deleted (called *hpi* mutant). This deletion removes the three main promoters of the HPI located around *ybtA*. We observed that the number and size of replicative foci were reduced with the *hpi* mutant strain (Figs. 4e and 5e), and the remaining foci did not present WGA labeling for exopolysaccharides around bacteria even if they have been phagocyted by the same macrophage as a WT LF82 (Fig. 5e). Viability in Raw 264.7, THP1, and HMDMs is affected by the deletion present in *hpi* mutant (Fig. 5f). Gallium salts are competitors of iron for siderophores, because of this property they have been successfully used as antibiotics against pathogenic bacteria[21]. As expected, the addition of $Ga^{3+}$ in the medium reduced the proportion of viable LF82 at 24 h of P.I. (Fig. 5g). Interestingly, we observed that the proportion of large IBCs increased in the presence of $Ga^{3+}$ suggesting that bacteria from IBC get a privileged access to iron or a protection from the toxic effects of $Ga^{3+}$ (Fig. 5h). Therefore, we postulate that yersiniabactin production might be higher inside IBCs compared to single cells.

**Regulation of iron capture by LF82 within macrophage.** To monitor HPI expression at the single cell level, we constructed a plasmid reporting the expression of the HPI promoter with an unstable GFP (P-HPI-GFP*). The expression of HPI started at 6 h of P.I., as expected from the RNA-seq data, and was evident at 24 h of P.I. (Fig. 6a). At 24 h of P.I., HPI expression was bimodal, with bacterial foci strongly expressing HPI while the other foci did not express HPI (Fig. 6a, b). Live imaging confirmed this bimodality with an expression starting in some IBC at 4 h and reaching a plateau at 10 h of P.I., while some other bacterial foci remained repressed for the entire kinetics (Supplementary Fig. 7). A similar induction kinetic was observed during the infection of THP1 macrophages (Supplementary Fig. 8). HPI expression was less frequent in the LF82 *csgD* (Fig. 6a, c), *adrA, bcsB, qseB* and *glgCAP* mutants (Fig. 6d), suggesting that the presence of the biofilm matrix is important for HPI expression. In *Yersinia pestis*, HPI expression is controlled by YbtA, a transcriptional activator, and the ferric uptake regulator Fur[22,23]. Fur also controls the expression of other iron capture systems such as the enterobactin, present in the core genome of enterobacteria. LF82 genome present 5 iron capture systems: the enterobactin system, the *sitABCD* system, the *Chu* haem capture system, a putative iron transpoter system and the HPI. Our RNA-seq experiments showed that inside macrophages, most iron capture systems were repressed, except the one encoded by HPI, which was highly overexpressed (Fig. 6e). This observation suggests that a particular regulation of the HPI is at play within IBCs and that the Yersiniabactin system, in particular, is adequate for iron

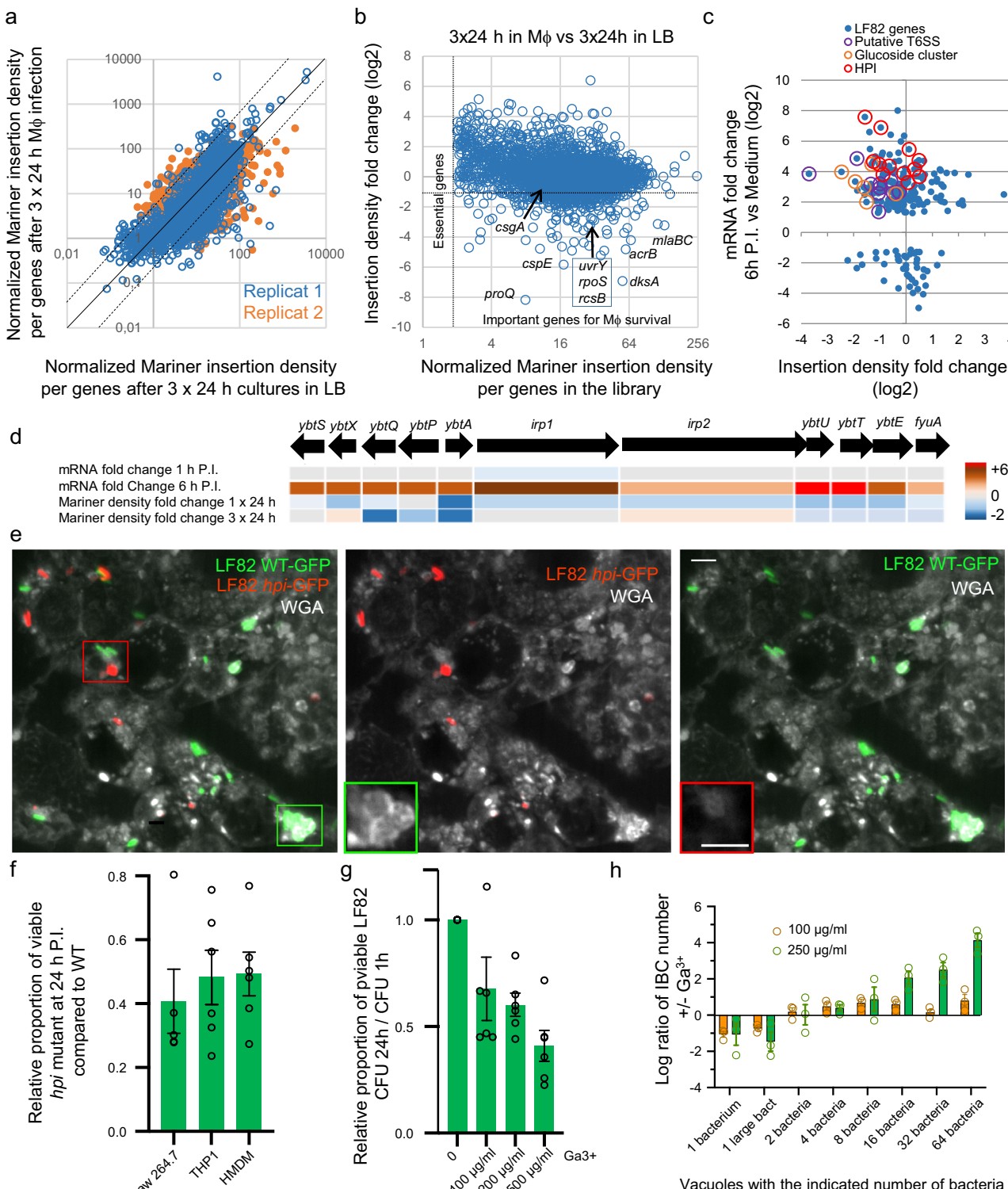

capture in the specific context of the IBCs of the AIEC LF82 strain within phagolysosomes.

**LF82 induces an iron response in the macrophage**. We explored human RNA-seq data for eventual signs of iron homeostasis dysregulation (Fig. 6f). Interestingly, the expression of several genes involved in iron homeostasis pathway was changed at 6 h of P.I. The IL6, ferritin, iron chaperones PCBP 1–3, and the ferrous iron transporter DMT1 were overexpressed, while we noticed a reduction in expression of

ferroportin, HCP1, HMOX1, and HMOX2. Surprisingly, we also noted that HAMP and LCN2 genes, encoding hepcidin and lipocalin, respectively, two of the main players of hypoferremia in response to infection by other pathogens, remained unaffected after infection (Supplementary Data 7). These data support a modification of iron homeostasis in the macrophage in correlation with the formation of LF82 IBCs, such synergy might reflect the expulsion of iron from phagolysosomes and its retention in ferritin within the cytoplasm or mitochondria of macrophages.

**Fig. 5 The high-pathogenicity island (HPI) contributes to IBC formation. a** Scatter plot representing Tn-seq replicates. **b** Scatter plot representing Tn-seq data. Genes presenting a high insertion density in the library (X axis) are under low selective pressure in LB and genes presenting low insertion density fold change (Y axis) are important for survival in the macrophage context. Selected important genes are annotated. **c** The RNA-seq and Tn-seq data of genes that are present in LF82 but not in K12 E. coli were collected (n = 814). This gene set contains many horizontally acquired genetic islands including pathogenicity island 1–4, metabolic gene islands, prophages, and many isolated genes of unknown function. We plotted RNA-seq data (mRNA fold change) according to TN-seq data (insertion density fold change) for this gene set. Only genes presenting a significant mRNA fold change (DE-seq P-value <$10^{-10}$) are represented. Genes from three pertinent islands (the HPI, a putative T6SS and a glucoside cluster) are highlighted. **d** Summary of RNA-seq and Tn-seq data for the HPI genes. **e** Imaging of LF82-GFP and LF82Δ*hpi* – mCherry co-infection 24 h of P.I. Images are Z projection of the maximum intensity of ten planes spaced by 400 nm. Insets correspond to the WGA labeling at the best focal plane for WT (green) and *hpi* mutant (red). Scale bars are 5 μm. **f** Proportion of viable bacteria at 24 h of P.I. in Raw, THP1, and HDMM macrophages in comparison to those present at 1 h. LF82 (green) and LF82 *hpi* deletion mutant (orange). **g** Proportion of viable bacteria at 24 h of P.I. in Raw, macrophages in comparison to those present at 1 h in the presence of $Ga^{3+}$ in the medium. Values represent the average of four experiments ± SEM. **h** Distribution of IBC's size in the population of the WT LF82 strain in the presence of $Ga^{3+}$. Values represent the average of four experiments ± SEM.

## Discussion

Although, there is a strong line of evidence supporting the role of AIEC in promoting gut inflammation and exacerbating CD pathology, the genetic determinants that discriminate AIEC from commensals have remained elusive. In the present work, we demonstrated that LF82 forms IBCs with biofilm-like traits inside phagolysosomes (Figs. 1 and 6g). Immediately upon infection, stringent response induction slowed the LF82 growth rate for a 6–10 h period. Late during this step, we observed the induction in bacteria of many genes involved in the biofilm pathway (Fig. 2). LF82 growth resumed concomitantly with the appearance of an extracellular matrix composed of exopolysaccharides and curli fibers (Fig. 3). Deletion (Fig. 4), interruption by a transposon of genes involved in the biofilm pathway (Fig. 5) or chemical inhibition by cis-2-decenoic acid (Fig. 4) stopped the formation of the exopolysaccharide matrix, and was detrimental for LF82 survival within murine and human macrophages (Fig. 4). Altogether, these results suggest that IBCs are an integral part of the adaptation that LF82 must undergo to survive within phagolysosomes. IBCs, however, are not the only survival mode of LF82 within macrophages; we observed, in every condition, isolated bacteria that survived, at least 24 h, inside macrophages. It is reasonable to propose that such isolated LF82 are persisters that did not resume growth in the time course of the infection. Future analysis will be required to evaluate this cell-to-cell heterogeneity and its contribution in the survival of intracellular LF82. IBCs with biofilm-like traits have also been observed for UPEC during invasion of the urothelial cells lining the urinary bladder but not inside vacuoles[10]. In this biofilm state, UPEC evade host defense and persist despite antibiotic therapy. However, the kinetics of UPEC's IBC formation differ radically from those observed for LF82; while UPEC biofilms contain non-growing bacteria, our observations suggest that AIEC's multiplication occurs inside the biofilm for IBC development (Fig. 3).

Biofilms formed on abiotic surfaces or on the epithelium are known to provide resistance to mechanical stress exerted by flow. At first glance, this might not be an important characteristic for IBCs that are confined within the phagolysosomal membrane. However, this characteristic might play a role in the long term if the macrophages die and release the IBCs. This structure might protect LF82 from dispersal by flow or from antibiotics present in the digestive tract and eventually from phagocytosis by another macrophage. In Crohn's disease patients, such a phenomenon might explain the chronic resurgence of AIEC[24]. Alternatively, it is possible that the biofilm acts as a physical barrier against the toxic compounds produced by macrophages, perhaps by limiting their diffusion into the core of the IBCs[25,26]. Finally, we can also postulate that IBCs provide a nutrient niche for LF82. The strong stringent response induction and the metabolic switches that we detected in the RNA-seq data suggest that phagolysosomal

activity leads to a severe depletion in nutrients for LF82. Cannibalism of dead LF82 might occur in the phagolysosome, and the matrix may facilitate this process[27]. We can postulate that in the context of the human gut, such multispecies IBCs are formed and that LF82 can feed on commensal bacteria. The presence of putative T6SSs in the genome of LF82 suggests that this strain is well armed to kill foreign IBC residents.

Using molecular genomic methods, we surveyed the genome of LF82 to identify determinants that allow this bacterium to form structured IBCs while commensal E. coli do not form such IBCs. LF82 harbors 925 genes that are absent from the genome of K12 MG1655. Among these genes, only a few seem to be specific to AIEC[28]; therefore, it is likely that the phenotypic characteristics of these bacteria emerged from a combination of genes or from pathoadaptive mutations[29]. Here, we present evidence suggesting that LF82 requires the combined action of the biofilm pathway encoded on the E. coli core genome and a particular iron capture system encoded by a pathogenicity island (HPI) acquired by horizontal transfer to establish the IBC. An analysis of the genetic diversity of E. coli isolated from patients with Crohn's disease that *irp2* prevalence was 70%[30]. In contrast, this association was only 30% for the control group. Enterobacteriaceae (Yersinia species or extra intestinal pathogenic E. coli (ExPEC)) that harbor HPI exhibit increased virulence. The role of HPI is to synthesize, export and capture Yersiniabactin bound to $Fe^{3+}$. Therefore, Yersiniabactin facilitates iron uptake, which is essential for bacterial growth. In an environment with very limited free iron, such as the human body, where most iron is bound to proteins, expression of the iron capture genes has been shown to be elevated[31]. We observed that 6 h of P.I., HPI was the only iron capture system with increased expression compared to the growth medium. This suggests that Yersiniabactin is particularly well suited for iron capture within macrophages. The chemical nature of the phagolysosome may limits the iron capture propensity of the enterobactin siderophore and other iron transport systems, increasing the selective pressure for Yersiniabactin. Finally, Yersiniabactin might also contribute to the protection of LF82 from iron[32] or copper[33] toxicity, which may be particularly relevant in phagolysosomal compartments and other restricted spaces in which reactive oxygen species are abundant. Our observations that a moderate dose of $Ga^{3+}$ simultaneously decreased the global viability of LF82 within macrophages, and increased the number of IBCs of large size support the hypothesis that IBCs create a more favorable environment than small vacuoles for LF82.

Our results show a direct link between iron capture and biofilm formation. We simultaneously observed that HPI expression is controlled by the biofilm pathway, and that the formation of the exopolysaccharide matrix is altered when HPI is deleted. This suggests a synergistic effect between these elements. Links between biofilms and iron homeostasis supporting this synergy

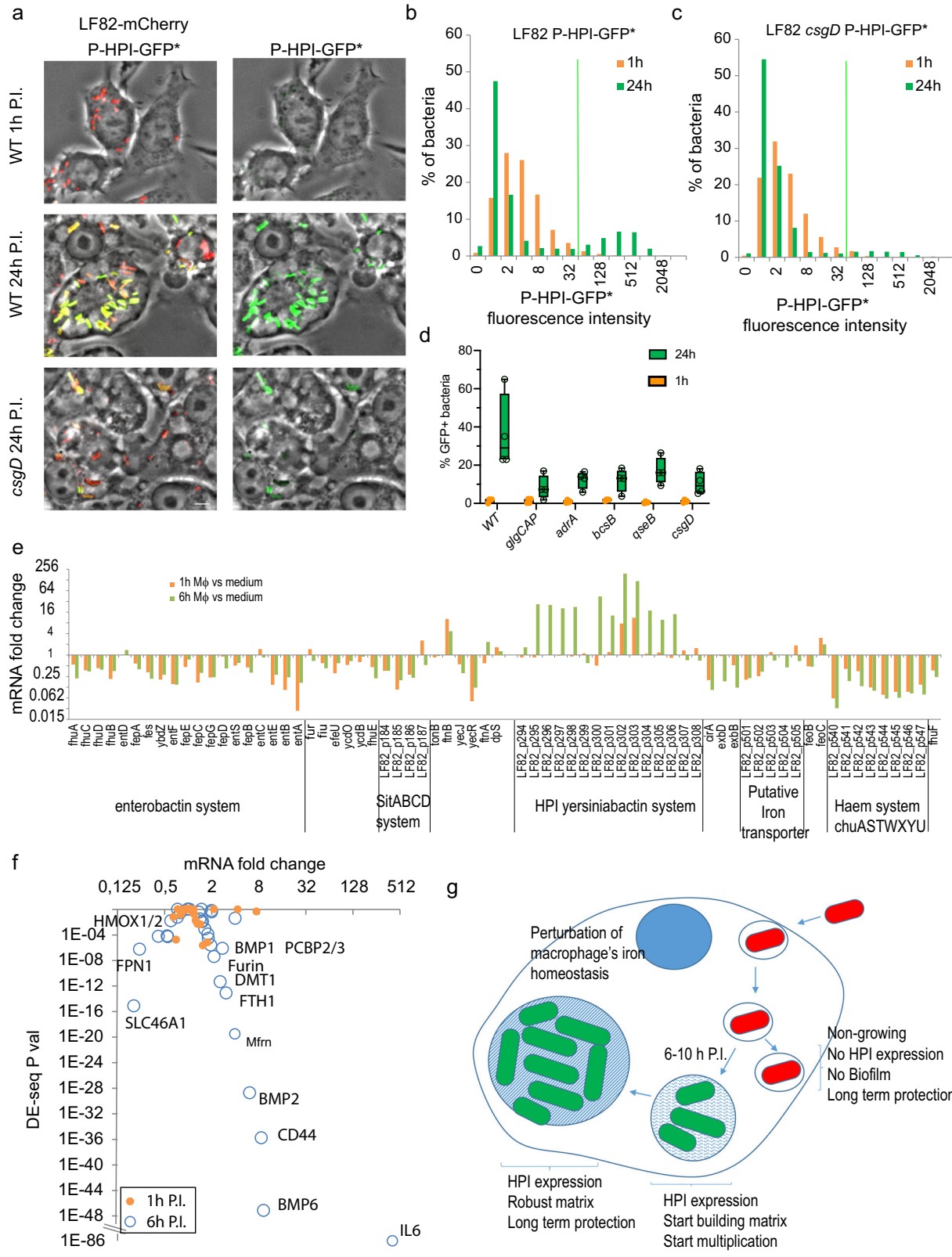

have already been documented in a few cases. First, it has been observed that the availability of iron greatly influences the ability of UPEC to form biofilms on the abiotic surface in urine medium (a poor source of iron); interestingly, this phenomenon was also dependent on the yersiniabactin system[34]. Second, *Pseudomonas aeruginosa* uses exopolysaccharides to sequester and store iron and stimulate biofilm formation[35,36]. Negatively charged exopolysaccharides chelate $Fe^{3+}$ and $Fe^{2+}$ in the vicinity of the bacteria and allow its capture. This scenario could be particularly interesting in the closed environment of the phagolysosomes, where iron might be expulsed by the DMT1 $Fe^{2+}$ transporter.

Altogether, our results suggest that for adapting to the harsh environment of the phagolysosomes (Fig. 6g), AIEC use a complex strategy involving a strong transcriptomic response, a

**Fig. 6 Exopolysaccharide matrix formation and iron homeostasis are interconnected. a** Imaging of the expression of the HPI promoter monitored with an unstable GFP fusion. WT and *csgD* mutant LF82–mCherry strains expressing P-HPI-GFP\* from a plasmid were imaged. Scale bars are 5 µm. **b** Quantification of the distribution of P-HPI-GFP\* fluorescence in LF82. The GFP fluorescence intensity of about 1000 m-Cherry objects was analyzed, data are from a representative experiment performed in triplicate. **c** Quantification of the distribution of P-HPI-GFP\* fluorescence in the LF82 *csgD* mutant. **d** Quantification of the % of P-HPI-GFP\*-positive cells in the WT strain and the *glgCAP, adrA, bcsB, qseB,* and *csgD* LF82 mutants. Values are the mean of three replicates, error bars are SD. **e** Expression data from RNA-seq experiments at 1 and 6 h of P.I. of THP1 macrophages for the genes annotated for their participation in iron homeostasis in LF82. Genes were organized according to their position on the genome. Clusters of genes are delineated by a vertical line. **f** Expression data from RNA-seq experiments for the genes annotated for their participation in iron homeostasis in humans. **g** Summary of our results unveiling LF82's strategy to survive within macrophage. Non-replicating AIEC (red), Replicating AIEC expressing HPI and the biofilm pathway (green), biofilm matix (blue waves), Nucleus (dark blue).

phenotypic switch, mechanisms to capture essential nutrients and a molecular shield to protect the IBC from environmental stress. The nutrient scarcity of the phagolysosomes, because it induces a stringent response and HPI expression, appeared to be an essential determinant of this strategy. Future work will be required to determine whether the phenotypic switch (growing–non-growing), production of the extracellular matrix and iron capture are connected pathways. Links between the stringent response and biofilm formation have already been documented in *Salmonella*[37]. Common regulators such as DksA, CsgD, and c-di-GMP for these three pathways might explain the kinetics of macrophage infection by LF82. Our observations suggest that other pathobiont bacteria equipped with a biofilm production apparatus, and an adequate iron capture system might also survive and form IBCs within macrophages. Does the colonization of this difficult niche represent a selective advantage for AIEC or other bacteria in the context of Crohn's disease or other diseases involving dysbiosis? Investigating this aspect should provide leads for future antibacterial-based strategies for precision medicine.

## Methods

**Infection, viable count, and confocal microscopy**. Cell lines and human macrophages derived from blood monocytes (HMDM) are described in the supplementary information. Macrophages were infected and imaged as described[18]. Antibiotic challenge and viable bacterial count was performed as described before[7]. Infections were performed at MOI 30 (measured by CFU), resulting in the observation of three LF82 bacteria per macrophage on average at 1 h of P.I. For control experiments with the K12-C600 strain infection were performed at MOI 200.

**FRAP**. FRAP was performed on Raw 264.7 macrophages infected with LF82-GFP. Infection was performed in fluorodish (World Precision Instruments). FRAP was performed at 24 h of P.I. Images were acquired using a Plan-APO 60×/1.4NA objective on a Ti Nikon microscope enclosed in a thermostatic chamber (Life Imaging Service) equipped with a Evolve EMCCD camera coupled to a Yokogawa CSU-X1 spinning disk. Metamorph Software (Universal Imaging) was used to collect data.

**STED imaging**. Infected macrophage were fixed and immunostained as described above. Imaging was performed with a STED expert Line Abberior Instruments GmbH coupled to a SliceScope Scientifica microscope. Images were acquired in two steps, first the confocal configuration for the mCherry bacteria (561 nm) and the Lamp1 staining (485 nm) and second with the STED configuration for the WGA staining (excitation at 640 nm, depletion at 775 nm). Detection was performed with avalanche photodiodes with filter cubes for green (500–550 nm), red (605–625 nm), and far red (650–720 nm) fluorescence.

**RNA-seq and Tn-seq**. Dual RNA-seq[12] and Tn-seq[19] experiments are described in the Supplementary information. The data are available at GEO # GSE154648.

**Statistics and reproducibility**. All data were expressed as mean and Standard Error of the Mean (SEM). *P* values were determined by the Student's unpaired *t*-test. The reproducibility was determined by using biological replicates and repeating the experiment 3–7 times as described in the figure legends.

**Reporting summary**. Further information on research design is available in the Nature Research Reporting Summary linked to this article.

## Data availability
Next-generation sequencing data has been deposited in GEO, Access numbers: GSE154648. All source data underlying the Tn-seq analysis are available in Supplementary Data 4 and 5. All source data underlying the graphs and plots are available in Supplementary Data 8.

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

## Acknowledgements

We thank the high throughput sequencing core facilities of I2BC (Centre de Recherche de Gif—http://www.i2bc.paris-saclay.fr/) and Curie Institute (supported by the ANR-10-EQPX-03 and ANR10-INBS-09-08 grants from the Agence Nationale de la Recherche and by the Canceropôle Ile-de-France). We are very grateful to Ugo Szachnowski, Nicolas Lapaque, Eric Allemand and the members of the CIRB imaging facility. This work has received support from ANR (https://anr.fr) with the references ANR-10-LABX-54 MEMOLIFE (O.E.) and ANR-11-IDEX-0001-02 PSL* Research University (O.E.), from the ANR with the reference ANR-18-CE35-0007 (O.E.), ANR-15-CE12-0007 (A.M.), from the European Research Council "EpincRNA" (starting) and "DARK" (consolidator) grants (A.M.) and the support of the association François Aupetit (AFA, https://www.afa.asso.fr) (OE).

## Author contributions

V.P., G.D., N.B., S.R., and O.E. designed the experiments. V.P., E.V., A.R., N.Q.D., J.D.G. performed and analyzed the infection experiments. V.P. performed Tn-seq experiments. G.D., M.A.B., M.W., E.V.D., and S.R. performed dual RNA-seq experiments. V.P., M.D., A.M., and O.E. analyzed RNA-seq data. V.P. and O.E. analyzed Tn-seq data. V.P. and O. E. wrote the manuscript.

## Competing interests

The authors declare no competing interests.
