## [Peer Review File · Communications Biology]

Reviewers' Comments:

Reviewer #1:

Remarks to the Author:

This is an interesting study addressing the molecular basis and functional consequences of intramacrophage biofilm formation by adherent invasive *E. coli* (AIEC). The authors combine cell culture infection assays with imaging, transcriptomics and functional genomics technologies to show that intracellular biofilms are composed of exopolysaccharides, their formation depends on the expression of a horizontally acquired pathogenicity island encoding the iron siderophore yersiniabactin, and biofilm production affects intracellular growth.

These are interesting findings and medically relevant given the association of AIEC with colonic disorders. I am, however, not sure if all conclusions are sufficiently supported by data and have a few technical concerns, as will be outlined below.

- Host cell model: Why are the authors switching back and forth between different macrophage cell lines for their different assays? E.g. mouse RAW cells are used for imaging (Fig. 1), while for the expression profiling they switch to human THP1 (Fig. 2), and for the Tn-seq go back to RAW (Fig. 5). This makes it very hard to cross-compare findings from different experiments. That the different cell lines can deliver vastly different results, is illustrated e.g. in Fig. 4 (compare the *glg* mutant between Fig. 4A and B).

- Dual RNA-seq: Please give more detail here as to how the assay was performed. E.g. I assume the authors did not enrich the invaded macrophages and separate them from the population of bystander cells. The host expression data thus unavoidably is the average expression over both populations; in order to better interpret it, the authors should clearly determine and show the relative proportion of invaded vs. bystander cells for both timepoints (1 h and 6 h p.i.). Further, there seems to be a high extent of cellular heterogeneity even within the population of invaded macrophages (see Fig. 6), which further complicates interpretation of the expression data (both, host and pathogen). These pitfalls – if not possible to circumvent them experimentally – should at least be clearly discussed in the manuscript.

Further, I was surprised that the authors depleted host rRNA prior to sequencing, but not that of the bacteria (see Method Appendix). Could the authors explain why this wasn't done. Also, in this case the mere number of bacterial reads per library (on average, 4 million; see page 4, line 17) is not useful. Instead, the authors should mention the average number of non-ribosomal bacterial reads per library.

Finally, I'd like to see some validation of the Dual RNA-seq data (e.g. the upregulation of HPI) on RNA and (ideally) the protein level.

- Also for the Tn-seq, I have some technical concerns: It is unclear to me, why the authors performed 3 rounds of infection. They state (page 7, line 29 ff.) that preliminary experiments suggested that one round of infection was not enough, but later (Fig. 5D) they show data from this 1-round infection assay and in fact, see stronger effects than for the 3-round experiment. Their assumption that these infection-associated effects might be compensated during the recovery phase (24 h in LB) should be better explained. How do the intracellular growth kinetics compare to the kinetics over 24 h under optimal conditions (LB)? That is, how often do AIEC duplicate during the infection assay and how often in LB?

Another concern I have is that the authors cannot discriminate effects on intracellular replication/survival and effects on initial uptake by macrophages. Looking at the genes whose disruption mutants were most strongly underrepresented in the output pool after infection, it seems as if host cell uptake is the primary selection pressure, rather than intracellular growth or survival.

Finally, and this refers to both techniques (Tn-seq and Dual RNA-seq), the respective methods are not well introduced and their introduction should be accompanied with citations to the respective key publications from the literature. Also, the global datasets should be uploaded on a public server (GEO or similar).

Reviewer #2:

Remarks to the Author:

In this manuscript, authors are tackling intra-macrophage behavior of an AIEC strain. Important to unravel the mechanisms of intramacrophage survival of this AIEC strain (LF82) because of its

association with Crohn's disease. Previous work has revealed a non-proliferative phase important because of the antibiotic recalcitrance it confers. Authors are now focusing on reporting and describing the proliferative phase of macrophage colonization by LF82. Interestingly, they observe a biofilm like structure within host cells, which is a novel strategy for survival. Authors use a battery of cutting-edge techniques, microscopy, FRAP, dual RNAseq and Tnseq to characterize thoroughly the interaction between LF82 and THP1 macrophages and identify the virulence factors that underpin such state. They uncover a pathogenicity island (HPI) that allows bacteria to acquire iron. The paper is well designed, straightforward and easy to follow. It is packed with important information and constitutes an essential piece of work to give a very complete overview of LF82 behavior in macrophages. This should be undoubtedly referenced on multiple occasions in the future.

Figure 1 – Authors make a compelling case for the fusion of vacuoles and the strong constraints imposed in the bacteria that do not move within vacuoles. The images are very convincing. Could the authors provide some quantification to support their observations? How many fused vacuoles (bicolor) did they observe? How many vacuoles did they photobleach?

Figure 2 – transcriptomic analyses of bacterial changes between 1 and 6h pi are convincingly supporting the hypothesis that LF82 is switching to a biofilm lifestyle during those first hours. However, panels C and D need reworking for legibility. I would recommend focusing on the key actors discussed in the text to simplify the graph and make it accessible to readers. These more extensive overviews presented now can be moved to supplementary data.

Figure 3 – Panel E is negative result that could be moved to supplementary data. Images are again very convincing, some quantification would strengthen the result presented in panels D and F.

Figure 4 – Have authors combined several mutations (e.g csgD and pgA) to see if survival defect is larger? Or do single mutant display already a total defect in extracellular matrix?

Figure 5 – Panel C would benefit from a more complete description of the approach used to combine RNAseq and Tnseq.

Minor:

P5I31 "Salmonella" capitalized and italicized.

P6L36 "is" missing

Figure 5A "replicate"

Reviewer #3:

Remarks to the Author:

The results of this work are of great interest for studying of phagocytosis and bacteria survival mechanisms. The data is extensive. The experiments are thoroughly and cleverly planned, and illustrations are colorful and clear. There is no doubt this manuscript deserves to be published in Communications biology.

There are, however, some minor points, that have to be addressed. More detailed methods description, addition to discussion and omitting some far-stretched statements will make this manuscript easier to read and understand.

Rebuttal for the manuscript **COMMSBIO-20-2682A. The Crohn's disease-related AIEC strain LF82 assembles a biofilm-like matrix to protect intracellular microcolonies from phagolysosomal attack**

Reviewer's comments are italicized in black and our responses are in blue

Reviewer #1

This is an interesting study addressing the molecular basis and functional consequences of intramacrophage biofilm formation by adherent invasive E. coli (AIEC). The authors combine cell culture infection assays with imaging, transcriptomics and functional genomics technologies to show that intracellular biofilms are composed of exopolysaccharides, their formation depends on the expression of a horizontally acquired pathogenicity island encoding the iron siderophore yersiniabactin, and biofilm production affects intracellular growth.

These are interesting findings and medically relevant given the association of AIEC with colonic disorders. I am, however, not sure if all conclusions are sufficiently supported by data and have a few technical concerns, as will be outlined below.

*• Host cell model: Why are the authors switching back and forth between different macrophage cell lines for their different assays? E.g. mouse RAW cells are used for imaging (Fig. 1), while for the expression profiling they switch to human THP1 (Fig. 2), and for the Tn-seq go back to RAW (Fig. 5). This makes it very hard to cross-compare findings from different experiments. That the different cell lines can deliver vastly different results, is illustrated e.g. in Fig. 4 (compare the *glg* mutant between Fig. 4A and B).*

We agree with the reviewer that using different cell lines make the reading of the manuscript more complex than if we kept to a single cell line. Our choice was driven by the observation that in THP1 macrophages individual IBCs contain less AIEC bacteria than in Raw 264.7 macrophages. Instead of the one large or two large vacuoles observed in Raw macrophages at 24h P.I., in THP1 we frequently observed multiple vacuoles containing a small number of bacteria (illustrated on the new SI S3C). Many of the experiments presented in the manuscript were therefore much more difficult in THP1. For example, co-infection events showing spatial clustering of LF82 (Figure 1B) impossible to observe in THP1. FRAP (Figure 1C) of small parts of the vacuoles was also impossible in THP1. Even within small vacuoles we unambiguously observed WGA labelling of the PGA matrix inside THP1 vacuoles (SI S3C). In the new supplementary figure SI S8B, we analyzed High Pathogenicity island expression in THP1 macrophages, it confirms our previous results in Raw macrophages. HPI is expressed in a subset of bacteria 5 -10 hours post infection and mark the starting point of IBC size increase.

To further homogenize our results, in this new version of the manuscript, we included new RNA-seq data performed with Raw 264.7 cells. Unfortunately this experiment did not function as well as the one performed with the THP1 cells. We obtained very few sequencing reads mapping of LF82 genome in this experiment ($\approx 0.05\%$ of bacteria reads and $>99\%$ of macrophage reads). It makes impossible to analyze the significance of this experiment. We have chosen to combine the reads of two duplicate experiments and to only analyze genes that were represented by at least 10 reads. The data corresponding of this gene's list are presented on the supplementary table S1, Log ratio comparison of THP1 and raw infection are in good correlation Pearson coefficient = 0.75 (SI S2B). Acid stress response, genotoxic stress, HPI and biofilm genes expression is overexpressed in these conditions (Table S1 and SI S2, SI S8). This is in very good correlation with our results in THP1. Since

they are expensive experiments, we cannot repeat RNA-seq experiment in Raw 264.7 cells for the moment. We also confirmed biofilm pathway and HPI expression by RtqPCR performed at different time point after Raw 264.7 infection (SI S2A).

• *Dual RNA-seq: Please give more detail here as to how the assay was performed. E.g. I assume the authors did not enrich the invaded macrophages and separate them from the population of bystander cells. The host expression data thus unavoidably is the average expression over both populations; in order to better interpret it, the authors should clearly determine and show the relative proportion of invaded vs. bystander cells for both timepoints (1 h and 6 h p.i.).*

The reviewer is right we did not enrich for invaded macrophages. The experiment is performed with an effective MOI of 3 at 1h post infection (ie an average of 3 bacteria per THP1 macrophage) in these conditions > 90% of the macrophages got infected, this number is not significantly changed in the first 10 hours of infection (Demarre et al 2019). Bystanders therefore contribute marginally to the transcriptomic signal. We have included these data in the supplementary text of the manuscript.

Further, there seems to be a high extent of cellular heterogeneity even within the population of invaded macrophages (see Fig. 6), which further complicates interpretation of the expression data (both, host and pathogen). These pitfalls – if not possible to circumvent them experimentally – should at least be clearly discussed in the manuscript.

We agree with the reviewer at late time point (> 10h P.I.) we observed an important heterogeneity within the population. This is well illustrated with the measure of IBC's size (Figure 4) and at the HPI expression level (Figure 6). The reviewer suggests analyzing this heterogeneity at the single cell level. Single cell transcriptomic is beyond the scope of the manuscript and beyond our technical capacities. Imaging data presented in the initial manuscript already described this heterogeneity; we have now included the following sentence in the discussion to emphasize on the importance of cell to cell heterogeneity for future work on IBCs.

“IBCs, however, are not the only survival mode of LF82 within macrophages; we observed, in every condition, isolated bacteria that survived, at least 24 h, inside macrophages. It is reasonable to propose that such isolated LF82 are persisters that did not resume growth in the time course of the infection. Future analysis will be required to evaluate this cell-to-cell heterogeneity and its contribution in the survival of intracellular LF82.”

Further, I was surprised that the authors depleted host rRNA prior to sequencing, but not that of the bacteria (see Method Appendix). Could the authors explain why this wasn't done. Also, in this case the mere number of bacterial reads per library (on average, 4 million; see page 4, line 17) is not useful. Instead, the authors should mention the average number of non-ribosomal bacterial reads per library.

We thank the reviewer; this was a mistake. We depleted for both eukaryotic and prokaryotic rRNA. It is now indicated in the method appendix.

Finally, I'd like to see some validation of the Dual RNA-seq data (e.g. the upregulation of HPI) on RNA and (ideally) the protein level.

Rt-qPCR and RNA-seq experiments (SI S2) were now also performed in Raw 264.7 macrophages, they confirm the expression of the biofilm pathway and the HPI. We did not success to detect *fyuA-3x* flag fusion in the macrophage. Since the amount of bacterial proteins is low compared to eukaryotic

proteins, this experiment is extremely difficult with pathogens that do not colonize extensively their host.

• Also for the Tn-seq, I have some technical concerns: It is unclear to me, why the authors performed 3 rounds of infection. They state (page 7, line 29 ff.) that preliminary experiments suggested that one round of infection was not enough, but later (Fig. 5D) they show data from this 1-round infection assay and in fact, see stronger effects than for the 3-round experiment. Their assumption that these infection-associated effects might be compensated during the recovery phase (24 h in LB) should be better explained. How do the intracellular growth kinetics compare to the kinetics over 24 h under optimal conditions (LB)? That is, how often do AIEC duplicate during the infection assay and how often in LB?

The referee is right this paragraph was unclear. We reformulated it and included a description of competition data that allowed us to plan the Tn-seq experiment: “Preliminary competitions experiments showed that the impact of known mutants affecting the growth phase of LF82 within macrophages was not easily observable after 24h of infection while it became obvious after 3 successive rounds of infection and LB outgrowth (SI S5B). “

Live microscopy experiments suggested that at the most LF82 divide 6-7 times in 24h. In LB, starting from an OD 0.01 (the starting dilution of the LB outgrowth for the Tn-seq control), LF82 also performs about 6-7 division to reach saturation around OD = 2.

Another concern I have is that the authors cannot discriminate effects on intracellular replication/survival and effects on initial uptake by macrophages. Looking at the genes whose disruption mutants were most strongly underrepresented in the output pool after infection, it seems as if host cell uptake is the primary selection pressure, rather than intracellular growth or survival.

The reviewer suggests that pathogen-associated molecular patterns (PAMPs) might appear in the Tn-seq results. Bacterial PAMPs are diverse; in gram negative bacteria they include formylated peptides and lipopolysaccharides. To test reviewer’s hypothesis, we analyzed the genes involved in LPS assembly. Tn-seq data shows that insertion inside many of these genes are positively selected after macrophage challenge (figure below). However, insertions frequency in these genes is low, suggesting, as expected, that they are detrimental in LB. Interpretation of the results obtained with this category of genes (low insertion frequency in the library) present a high risk of artefacts. We did not chose to continue on this lead.

Finally, and this refers to both techniques (Tn-seq and Dual RNA-seq), the respective methods are not well introduced and their introduction should be accompanied with citations to the respective key publications from the literature. Also, the global datasets should be uploaded on a public server (GEO or similar).

We have revised the introduction of RNA-seq and Tn-seq and included new references.

The data are available at GEO # GSE154648, this is indicated in the method section.

Reviewer #2 (Remarks to the Author):

In this manuscript, authors are tackling intra-macrophage behavior of an AIEC strain. Important to unravel the mechanisms of intramacrophage survival of this AIEC strain (LF82) because of its association with Crohn's disease. Previous work has revealed a non-proliferative phase important because of the antibiotic recalcitrance it confers. Authors are now focusing on reporting and describing the proliferative phase of macrophage colonization by LF82. Interestingly, they observe a biofilm like structure within host cells, which is a novel strategy for survival. Authors use a battery of cutting-edge techniques, microscopy, FRAP, dual RNAseq and Tnseq to characterize thoroughly the interaction between LF82 and THP1 macrophages and identify the virulence factors that underpin such state. They uncover a pathogenicity island (HPI) that allows bacteria to acquire iron. The paper is well designed, straightforward and easy to follow. It is packed with important information and constitutes an essential piece of work to give a very complete overview of LF82 behavior in macrophages. This should be undoubtedly referenced on multiple occasions in the future.

Figure 1 – Authors make a compelling case for the fusion of vacuoles and the strong constraints imposed in the bacteria that do not move within vacuoles. The images are very convincing. Could the

authors provide some quantification to support their observations? How many fused vacuoles (bicolor) did they observe? How many vacuoles did they photobleach?

The quantification of color sectors after fusion of the vacuoles turned to be impossible; it might require 3D segmentation tools that we do not have in hands. We have changed figure 1 to include average values +/- SD of the FRAP experiments (N=10)

Figure 2 – transcriptomic analyses of bacterial changes between 1 and 6h pi are convincingly supporting the hypothesis that LF82 is switching to a biofilm lifestyle during those first hours. However, panels C and D need reworking for legibility. I would recommend focusing on the key actors discussed in the text to simplify the graph and make it accessible to readers. These more extensive overviews presented now can be moved to supplementary data.

As suggested by the reviewer we transferred the box plot on Figure 2C to the supplementary data and made a simpler figure with the main pathway modified at 6h P.I. We did not change figure 2D as it is. We think it is important to show that biofilm matrix production is original inside IBC with curli, PGA but not cellulose.

Figure 3 – Panel E is negative result that could be moved to supplementary data. Images are again very convincing, some quantification would strengthen the result presented in panels D and F.

Panel E was removed from figure 3 and a quantification of the curli signal included in the panel F

Figure 4 – Have authors combined several mutations (e.g csgD and pgA) to see if survival defect is larger? Or do single mutant display already a total defect in extracellular matrix?

No, we did not combine mutations yet. This is a good idea. Because of natural antibiotic resistance genes in the genome of LF82 and the poor efficiency of recombineering protocol the construction of double mutants is difficult. We avoid them when possible.

Figure 5 – Panel C would benefit from a more complete description of the approach used to combine RNAseq and Tnseq.

We modified the legend as follow:

C) The RNA-seq and Tn-seq data of genes that are present in LF82 but not in K12 E. coli were collected (n= 814). This gene set contains many horizontally acquired genetic islands including pathogenicity island 1 to 4, metabolic gene islands, prophages and many isolated genes of unknown function. We plotted RNA-seq data (mRNA fold change) according to TN-seq data (insertion density fold change) for this gene set. Only genes presenting a significant mRNA fold change (DE-seq P-value <10⁻¹⁰) are represented. Genes from 3 pertinent islands (the HPI, a putative T6SS and a glucoside cluster) are highlighted.

Minor:

P5I31 “Salmonella” capitalized and italicized.

P6L36 “is” missing

Figure 5A “replicate”

These point were corrected

Reviewer #3 (Remarks to the Author):

The results of this work are of great interest for studying of phagocytosis and bacteria survival mechanisms. The data is extensive. The experiments are thoroughly and cleverly planned, and illustrations are colorful and clear. There is no doubt this manuscript deserves to be published in Communications biology.

There are, however, some minor points, that have to be addressed. More detailed methods description, addition to discussion and omitting some far-stretched statements will make this manuscript easier to read and understand.

We thank the reviewer for his enthusiasm, we have modified the method section and the discussion in many aspects (see above). These modifications should help reading as suggested by reviewer#3

Reviewers' Comments:

Reviewer #1:

Remarks to the Author:

I still think this is a timely and interesting piece of work. In their revised version, the authors tried to address my previous comments.

- With respect to the host cell model, I previously raised the comment that switching back and forth between human and mouse macrophage cell lines makes it very hard to cross-compare findings from different experiments. The authors replied, explaining that this was due to technical challenges when working with THP1 cells. "Many of the experiments presented in this manuscript were therefore much more difficult in THP1." In an effort to provide some additional experimental data to illustrate correlation of their findings between macrophage models, for their revised manuscript the authors performed an RNA-seq experiment in Raw cells. Somewhat counterintuitively in light of their above statement, the transcriptome profiling – which had worked well with THP1 – "did not function as well" in RAW macrophages.

- Regarding the Tn-seq, I had previously raised technical concerns because the authors cannot discriminate effects on intracellular replication/survival and effects on initial uptake by macrophages. Indeed, the re-analysis of their data that was triggered by this comment revealed "that insertion inside many of these genes are positively selected after macrophage challenge". However, since insertion frequency was generally low for these genes they "did not chose to continue on this lead." I still believe that this technical limitation of their experimental design should at least be clearly communicated in the manuscript.

- The last comment I made previously referred to the Tn-seq and Dual RNA-seq techniques, which were not well introduced and cited. Here again, I still think that when "dual RNA-seq" and "TN-seq" are first mentioned in the text (line 62), they should be accompanied with proper citations to at least each one comprehensive review, which exist for both techniques. Currently, this is limited to the Methods and Supplementary sections.

On a more general note: From a reviewer's standpoint it would be very much appreciated if text changes made during revision were marked (e.g in a different color).

Reviewer #2:

Remarks to the Author:

Authors have thoroughly addressed all of my comments on the first version of their manuscript. They also addressed very seriously all the other points raised by the other reviewers and even went as far as redoing some RNAseq to unify their experiments and infection model. The manuscript is in great shape and deserves to be published.

Reviewer #1 (Remarks to the Author):

I still think this is a timely and interesting piece of work. In their revised version, the authors tried to address my previous comments.

Q: With respect to the host cell model, I previously raised the comment that switching back and forth between human and mouse macrophage cell lines makes it very hard to cross-compare findings from different experiments. The authors replied, explaining that this was due to technical challenges when working with THP1 cells. “Many of the experiments presented in this manuscript were therefore much more difficult in THP1.” In an effort to provide some additional experimental data to illustrate correlation of their findings between macrophage models, for their revised manuscript the authors performed an RNA-seq experiment in Raw cells. Somewhat counterintuitively in light of their above statement, the transcriptome profiling – which had worked well with THP1 – “did not function as well” in RAW macrophages.

A: The reviewer appreciated our efforts to reply is comment

Q: Regarding the Tn-seq, I had previously raised technical concerns because the authors cannot discriminate effects on intracellular replication/survival and effects on initial uptake by macrophages. Indeed, the re-analysis of their data that was triggered by this comment revealed “that insertion inside many of these genes are positively selected after macrophage challenge”. However, since insertion frequency was generally low for these genes they “did not chose to continue on this lead.” I still believe that this technical limitation of their experimental design should at least be clearly communicated in the manuscript.

A: We have to disagree with the reviewer. The low frequency of transposon insertions in LPS genes reveals that they are essential or very detrimental in the growth medium. Their analysis in the macrophage Tn-seq data will only produce artefacts. It is not useful to comment on them in the manuscript.

Q: The last comment I made previously referred to the Tn-seq and Dual RNA-seq techniques, which were not well introduced and cited. Here again, I still think that when “dual RNA-seq” and “TN-seq” are first mentioned in the text (line 62), they should be accompanied with proper citations to at least each one comprehensive review, which exist for both techniques. Currently, this is limited to the Methods and Supplementary sections.

A: We have included references to TN-seq and RNAseq in the result section.

Reviewer #2 (Remarks to the Author):

Authors have thoroughly addressed all of my comments on the first version of their manuscript. They also addressed very seriously all the other points raised by the other reviewers and even went as far as redoing some RNAseq to unify their experiments and infection model.

The manuscript is in great shape and deserves to be published.

We thank the reviewer for his enthusiasm.